# Chemoresistant Cancer Cell Lines Are Characterized by Migratory, Amino Acid Metabolism, Protein Catabolism and IFN1 Signalling Perturbations

**DOI:** 10.3390/cancers14112763

**Published:** 2022-06-02

**Authors:** Mitchell Acland, Noor A. Lokman, Clifford Young, Dovile Anderson, Mark Condina, Chris Desire, Tannith M. Noye, Wanqi Wang, Carmela Ricciardelli, Darren J. Creek, Martin K. Oehler, Peter Hoffmann, Manuela Klingler-Hoffmann

**Affiliations:** 1Clinical & Health Sciences, University of South Australia, Adelaide, SA 5095, Australia; mitch.acland@gmail.com (M.A.); clifford.young@unisa.edu.au (C.Y.); mark@massdynamics.com (M.C.); chris.desire@unisa.edu.au (C.D.); oehler.mk@gmail.com (M.K.O.); peter.hoffmann@unisa.edu.au (P.H.); 2Adelaide Proteomics Centre, School of Biological Sciences, The University of Adelaide, Adelaide, SA 5000, Australia; 3Discipline of Obstetrics and Gynaecology, Adelaide Medical School, Robinson Research Institute, The University of Adelaide, Adelaide, SA 5005, Australia; noor.lokman@adelaide.edu.au (N.A.L.); tannith.noye@adelaide.edu.au (T.M.N.); wanqi.wang@adelaide.edu.au (W.W.); carmela.ricciardelli@adelaide.edu.au (C.R.); 4Monash Institute of Pharmaceutical Sciences, Monash University, Parkville, VIC 3052, Australia; dovile.anderson@monash.edu (D.A.); darren.creek@monash.edu (D.J.C.); 5Department of Gynaecological Oncology, Royal Adelaide Hospital, Adelaide, SA 5005, Australia

**Keywords:** ovarian cancer, chemoresistance, cancer cell lines, proteomics, metabolomics

## Abstract

**Simple Summary:**

While chemoresistance remains a major barrier to improving the outcomes for patients with ovarian cancer, the molecular features, and associated biological functions, which underpin chemoresistance in ovarian cancer remain poorly understood. In this study we aimed to provide insight into the proteins and metabolites, and their associated biological pathways, which play a role in conferring chemoresistance to ovarian cancer. Through mass spectrometry analysis comparing the proteome and metabolome of chemosensitive vs chemoresistant ovarian cancer cell lines we revealed numerous perturbations in signalling and metabolic pathways in chemoresistant cells. Further comparison to primary cells taken from patients with chemoresistant or chemosensitive disease identified a shared dysregulation in cytokine and type 1 interferon signalling. Our research sets the foundation for a deeper understanding of the proteomic and metabolomic features of chemoresistance and identifies type 1 interferon signalling as a common feature of chemoresistance.

**Abstract:**

Chemoresistance remains the major barrier to effective ovarian cancer treatment. The molecular features and associated biological functions of this phenotype remain poorly understood. We developed carboplatin-resistant cell line models using OVCAR5 and CaOV3 cell lines with the aim of identifying chemoresistance-specific molecular features. Chemotaxis and CAM invasion assays revealed enhanced migratory and invasive potential in OVCAR5-resistant, compared to parental cell lines. Mass spectrometry analysis was used to analyse the metabolome and proteome of these cell lines, and was able to separate these populations based on their molecular features. It revealed signalling and metabolic perturbations in the chemoresistant cell lines. A comparison with the proteome of patient-derived primary ovarian cancer cells grown in culture showed a shared dysregulation of cytokine and type 1 interferon signalling, potentially revealing a common molecular feature of chemoresistance. A comprehensive analysis of a larger patient cohort, including advanced in vitro and in vivo models, promises to assist with better understanding the molecular mechanisms of chemoresistance and the associated enhancement of migration and invasion.

## 1. Introduction

Ovarian cancer (OC) is the deadliest gynaecological malignancy, with survival rates of below 30% when detected at a late stage [1]. Owing to its often asymptomatic progression, 70% of ovarian cancers are diagnosed at a late stage [2]. The 5-year survival rate for a patient diagnosed with advanced ovarian cancer is approximately 35% [1], and this has not increased substantially over the last 30 years [3].

The typical treatment regime for ovarian cancer consists of debulking surgery, followed by platinum-based chemotherapy with carboplatin (CBP) [4]. It is estimated that 20% of patients do not respond to chemotherapy due to the innate chemoresistance of their OC, while 70% of patients experience acquired chemoresistance [5]. High-grade serous ovarian carcinoma (HGSOC) represents the majority of advanced OC, and is characterized by an initial response to treatment, followed by recurrence and the development of chemoresistance [6,7], and is responsible for the majority of ovarian cancer-associated fatalities [8]. This highlights chemoresistance as being the main barrier to improving survival rates for this deadly cancer.

Despite the development of numerous novel anti-cancer agents against OC [9], platinum-based chemotherapy drugs remain the first-line treatment of choice [4]. They are the National Comprehensive Cancer Network-recommended treatment following surgery for all Stage II, III, and IV ovarian cancers [10]. These drugs act primarily through the formation of intra- and inter-strand DNA cross-links, which induce cell cycle arrest, typically at the G2/M checkpoint [11].

In addition, platinum-based chemotherapeutics also contribute to increased oxidative stress within the cell, through the production of reactive oxygen species (ROS) [12]. In cancer cells, where greater oxidative stress is exhibited in comparison with normal cells, carboplatin-induced oxidative stress can result in apoptosis [13].

The DNA damage and oxidative stress induced by platinum chemotherapy triggers a complex web of signalling, which both promotes and inhibits cell death. This includes signalling through MAPK, PI3K, JNK and other pathways (reviewed in [12]). In addition, platinum chemotherapies have broader impacts on protein folding [14] and calcium homeostasis [14], and they inhibit the function of certain proteins, influencing transcription [15] and microtubule formation [16].

Cisplatin has historically been the most commonly used platinum-based chemotherapy, and it has been employed extensively to treat a range of cancers since its approval in 1978 [17]. However, cisplatin has significant side effects that limit the concentration that can be used for treatment [18]. The chemical analogue, carboplatin, has significantly less side effects and a similar efficacy when combined with paclitaxel, and it is now the preferred first-line treatment of ovarian cancer [19].

There are several mechanisms through which resistance to platinum-based chemotherapy can occur, including the impaired influx of the compound, increased drug efflux, cytoplasmic detoxification, increased DNA repair and alterations in apoptosis signalling pathways [20]. The DNA crosslinks formed by carboplatin give rise to incomplete DNA repair, resulting in single- and double-stranded breaks giving rise to apoptotic signalling [21]. DNA repair mechanisms and apoptosis signalling pathways are of great importance in the resistance to platinum-based chemotherapy.

There has been significant research into the genetic and molecular underpinnings of HGSOC, which has provided a deeper understanding of this cancer and its progression. Though these efforts have revealed the deep complexity and heterogeneity of HGSOC, they have failed to translate into improved clinical outcomes. It has been long understood that altered metabolism is a key feature of cancer cells that is best characterised by the Warburg effect, where cancer cells favour glycolysis over oxidative phosphorylation [22]. Metabolism contributes to a vast array of cancer biological features, including oncogenic transformation, growth, stress response and the detoxification of damaging chemical agents [23]. It has been demonstrated that the downregulation of glycolytic enzymes has the potential to overcome chemoresistance [24,25]; however, the mechanisms are not well understood.

One way to investigate the molecules that underpin chemoresistance is through the application of mass spectrometry (MS) coupled with high-performance chromatography techniques. This approach can be applied to identify a range of molecules including metabolites and proteins. Through the application of this technique, hundreds of metabolites and thousands of proteins can be routinely identified and relatively quantified from very small amounts of material.

To provide a comprehensive molecular characterisation of chemoresistant HGSOC, we combined MS-based metabolomic and proteomic analyses of these cell lines. Together, this holds the potential to provide a deep characterisation of these cell lines, and to reveal alterations in proteomic and metabolomic pathways that underpin chemoresistance in this model.

## 2. Materials and Methods

### 2.1. Cell Culture

The human OC cell line CaOV3 was purchased from the American Type Culture Collection (ATCC, Manassas, VA, USA), and the OVCAR-5 cell line was obtained from Dr. Thomas Hamilton (Fox Chase Cancer Centre, Philadelphia, PA, USA). Both cell lines were authenticated via a short tandem repeat (STR) DNA profile in 2020. The OVCAR-5 cells were grown in RPMI 1640 media (Sigma Aldrich, St. Louis, MO, USA). Recent reports have indicated that OVAR-5 might originate from metastatic gastrointestinal cancer, and were potentially wrongfully labelled as being ovarian cancer [26]. CaOV3 cells were grown in DMEM media (Sigma Aldrich, St. Louis, MO, USA). Both cell lines were cultured with the addition of 10% foetal bovine serum (Bovogen Biologicals, East Keilor, VIC, Australia) supplemented with 1% penicillin/streptomycin (Sigma Aldrich, St. Louis, MO, USA) and 1% L-glutamine (Sigma Aldrich, St. Louis, MO, USA). OVCAR-5 and CaOV3 cells were made resistant to CBP after treatment with 6–8 cycles of CBP (50 μM, Hospira Australia, Pty Ltd., Mulgrave, VIC, Australia) [27,28]. Resistance to CBP was measured regularly, and CBPR cell lines were seen to be at least two-fold more resistant to CBP than their parental partners through the following experiments.

### 2.2. Primary HGSOC Culture

Primary cells were isolated from the ascites of advanced stage HGSOC patients (*n* = 2), with patient consent and approval, by the Royal Adelaide Hospital RAH and Central Adelaide Local Health Network Human Ethics Committees (RAH # 140201) (CALHN # R20181215). All primary cells were grown in Advanced RPMI 1640 medium (Life Technologies, Mulgrave, VIC, Australia) supplemented with 4 mM L-glutamine, 10% FBS (Sigma Aldrich, St. Louis, MO, USA), and antibiotics (100 U penicillin G, 100 µg/mL streptomycin sulphate, and 100 µg/mL amphotericin B, Sigma Aldrich). Patients that experienced a recurrence within six months after finishing first-line CBP+paclitaxel chemotherapy were classified as being chemoresistant. Patients that remained in full remission for longer than 6 months were classified as being chemosensitive. Here, we investigated the proteome of cells derived from one patient who was deemed chemoresistant and one who was deemed chemosensitive using the above-mentioned classifications. Patient diagnoses and chemotherapy responses are outlined in Appendix A.

### 2.3. In Vitro Motility Assay

OVCAR-5 cell motility was assessed using a ChemoTx^®^ 96-well plate (Neuroprobe, Gaithersburg, MD, USA), as previously described [29]. Briefly, cells were labelled with calcein AM (1 µg/mL, Life Technologies, Mulgrave, VIC, Australia) for 30 min in the dark on a nutator. Excess calcein AM was removed by washing with media (RPMI1640 + 0.1% BSA). A concentration of 40,000 cells/50 µL were loaded onto uncoated 12 µm filter inserts. The cells were allowed to migrate for 6 h to the bottom well with chemoattractant (10% FBS) and media (RPMI 1640 + 0.1% BSA). Migrated cells were measured using the Triad series multimode detector (Dynex Technologies, Chantilly, VA, USA) at 485–520 nm. Assays were carried out in biological quadruplicate in 2–3 independent experiments.

### 2.4. Chick Chorioallantoic Membrane (CAM) Assay

The CAM assay was performed as previously described [30]. In brief, 90,000 OVCAR-5 cells were mixed with Matrigel (8.9 mg/mL, BD Biosciences, Melbourne, VIC, Australia) and placed on top of the CAM of Day 11 chick embryos. Matrigel grafts with adjacent CAM were harvested from each embryo (*n* = 6–9/treatment group) after 3 days (day 14), fixed with 10% formalin for 24 h, processed, and embedded in paraffin. Serial sections (6 μm) were stained with haematoxylin and eosin, or immunostained with the monoclonal mouse anti-human cytokeratin clone AE1/AE3 (1:50 Dako-Agilent Santa Clara, CA, USA). Immunohistochemistry was performed as described previously, using citrate buffer antigen retrieval [29]. Slides were digitally scanned using the NanoZoomer (Hamamatsu Photonics). A quantitative analysis for assessing OVCAR-5 cancer cell invasion was performed on 8 to 12 CAM images for each embryo, as previously described [30].

### 2.5. Cell Survival Assay

Ovarian cancer cells were plated at 5000 cells/well in 96-well plates in the respective growth media. After 24 h, cells were treated with varying concentrations of CBP (0–200 µM) for 72 h. Then, the conditioned media was removed and thiazolyl blue tetrazolium bromide (MTT) (0.5 mg/mL, Sigma Aldrich) was added for 4.5 h, followed by MTT solvent (0.1 N HCl in isopropanol) for 10 min, before absorbance readings were measured at 595 nm using a Triad series multimode detector plate reader (Dynex technologies, Chantilly, VA, USA). The CBP IC50 values were calculated using a non-linear fit from the variable slope of log (inhibitor), using GraphPad Prism.

### 2.6. Metabolomics Sample Preparation

Cells were maintained at 60–80% confluence for 3 passages before being plated in 10 cm dishes. Cell numbers were estimated from an additional dish with the same number of cells at seeding. The media was aspirated and the cells were washed 3 times with 3 mL warmed PBS. Metabolic arrest was achieved through the addition of approximately 2 mL of liquid nitrogen directly to cells ensuring that the surface of the plate was covered before plates were placed onto dry ice. Metabolite extraction was achieved through the addition of 1 mL 100% ice-cold methanol. Cells were lifted off the plate using an ice-cold cell scraper and transferred to a 2 mL Eppendorf. An additional 1 mL of 100% ice-cold methanol was added before snap freezing via immersion in liquid nitrogen for 3 min. This was followed by thawing on dry ice and vortexing to resuspend the contents. The freeze/thaw process was repeated 5 times to ensure the full extraction of the metabolites. The samples were centrifuged at 16,000× *g* at −9 °C for 5 min, and the supernatant was retained. The pellet was resuspended in 500 uL of 100% ice-cold methanol and freeze/thawed 5 times in liquid nitrogen. This sample was centrifuged at 16,000× *g* at −9 °C for 5 min, and the supernatant was retained and combined with the previously retained supernatant. The samples were then dried in a SpeedVac Vacuum Concentrator (John Morris Scientific, RVC 2-18) at room temperature, with a vacuum of 40 mbar and a rotor speed of 1000 min^−1^. Before data acquisition, the samples were resuspended in volumes of 20 mM ammonium carbonate and acetonitrile to achieve an identical concentration of biological material based on the cell number estimate.

### 2.7. Metabolomics Data Acquisition

LCMS data was acquired on a Q-Exactive Orbitrap mass spectrometer (Thermo Fisher) coupled with a Dionex Ultimate^®^ 3000 RS high-performance liquid chromatography (HPLC) system (Thermo Fisher). Chromatographic separation was performed on a ZIC-pHILIC column (5 µm, polymeric, 150 × 4.6 mm, SeQuant^®^, Merck, Darmstadt, Germany). The mobile phase was (A) 20 mM ammonium carbonate and (B) acetonitrile. The gradient program started at 80% B and was reduced to 50% B over 15 min, then this reduced from 50% B to 5% B over 3 min, followed by a wash with 5% B for another 3 min, and finally an 8 min re-equilibration with 80% B. The flow rate was 0.3 mL/min and the column compartment temperature was 25 °C. The total run time was 32 min with an injection sample volume of 10 µL. The mass spectrometer operated in full scan mode with positive and negative polarity switching at 35,000 resolution and 200 *m/z*, with detection range of 85 to 1, 275 *m/z* in full scan mode. The electro-spray ionisation source (HESI) was set to 3.5 kV for the positive mode and 4.0 kV for the negative mode, the sheath gas was set to 50 and the aux gas to 20 arbitrary units, the capillary temperature was 300 °C and the probe heater temperature was 120 °C.

Mixtures of pure authentic standards containing over 320 metabolites were acquired as separate injections and used to confirm the retention times. The metabolites confirmed with the authentic standards were given the highest confidence MSI level 1.

The metabolomics data were deposited in the data repository Metabolomics Workbench [31] under the study ID ST002010 (DOI: http://dx.doi.org/10.21228/M81Q4Z, accessed on 21 April 2022).

### 2.8. Metabolomics Data Analysis

The acquired LCMS data was processed in an untargeted fashion using the open-source software IDEOM [32,33]. Default IDEOM parameters were used to eliminate unwanted noise and artefact peaks. The putative identification of metabolites was achieved using accurate mass (within 3 ppm mass error) searching against the Kyoto Encyclopedia of Genes and Genomes (KEGG), MetaCyc, and LIPIDMAPS databases and others.

Despite the washing steps performed in sample preparation, it is expected that some metabolites that were present in the cell culture media may influence the metabolite abundances observed in our cell samples. To avoid their influence on our results, we utilized a media-only blank performing the same sample preparation steps on a plate with no cells and only 10 mL of cell culture media. To exclude the media components from our analysis, we excluded all metabolites where the media blank/control ratio was greater than 0.5, for the downstream analysis.

### 2.9. Metabolomic Functional Pathway Analysis

To understand the functional pathways associated with dysregulated metabolites, we utilized the Enrichment Analysis function on the Metaboanalyst platform [34,35] using the default settings, and comparing them to the SMPDB metabolomics database.

### 2.10. Cell Lysis and Acetone Precipitation

Cells pellets containing up to 1 × 10^7^ cells were collected and washed three times with PBS before being stored at −80 °C for lysis. Pellets were lysed on ice via resuspension in 200 uL RIPA buffer (Reference Number: 20–188, Millipore) supplemented with 1% (*v*/*v*) protease inhibitor cocktail (Reference Number: P8340, Sigma Aldrich, St. Louis, MO, USA). The solution was then passed through a 26.5 G needle (reference number: NN+2613R, Terumo) 5 times before being spun at 20,000 G for 30 min in a centrifuge pre-cooled to 4 °C. The supernatant was transferred to a second tube and 4 volumes of ice-cold acetone were added. The tube was then gently mixed and incubated overnight at −20 °C. Using a centrifuge precooled to −9 °C, samples were spun at 20,000 G for 10 min and the supernatant was carefully removed. The pellet was then washed twice with 200 uL of ice-cold acetone to ensure the removal of any remaining contaminants. Finally, the pellet was air dried on ice for 20 min or until all liquid had evaporated. The pellet was then dissolved in 8 M urea with 50 mM ammonium bicarbonate before the protein content was estimated using a tryptophan fluorescence assay.

### 2.11. Tryptophan Fluorescence Assay for Protein Estimation

Protein quantification was performed using tryptophan fluorescence [36]. Briefly, 10 μL of sample was diluted in 90 μL of 8 M urea (1:10) before 50 µL was added to a 40 µL fluorescence cell (Agilent Technologies, P.N. 6610021600). The fluorescence was read using an Agilent Carry Eclipse Fluorescence Spectrophotometer G9800A (Agilent Technologies, S.N. MY13260001) with software version 1.2 (146), applying the following settings: excitation = 295 nm (bandwidth = 5 nm), emission = 350 nm (bandwidth = 20 nm), average time = 0.1 s and PMT starting voltage = 560 V. The cell was washed with water and 8 M urea before measuring the next sample. Fluorescence readings were compared to a 9-point L-tryptophan (Sigma-Aldrich, ≥98%, P.N. T0254) standard curve (9.15 × 10^−5^ to 0.0117 mg/mL) and multiplied by a conversion factor (85.47) determined for the average number of tryptophan residues in mammalian proteins.

### 2.12. Protein Digestion and Clean Up

A total of 100 µg of purified protein in 100 µL of 8 M urea (Merck, Kenilworth, NJ, USA) with 50 mM ammonium bicarbonate (pH 8.0) (Fluka Analytical, P.N. 09830) was reduced via the addition of 10 mM DTT (Sigma-Aldrich) and incubated at room temperature for 1 h. Samples were then reduced via the addition of 15 mM chloroacetamide (CA) and incubated in the dark at room temperature for 30 min. Samples were diluted with 900 uL of 50 mM (pH 8.0) before the addition of 2 µg of trypsin/Lys-C (Promega) (1:50 trypsin: protein) and incubation for 8 h at 37 °C.

Samples were then cleaned up using a C18 Sep-Pak (Waters) equipped to a vacuum manifold. The Sep-Pack was washed with 1 mL methanol before being equilibrated via the addition of 1 mL 80% acetonitrile with 0.1% formic acid (FA) 3 times, followed by the addition of 1 mL of 0.1% FA, 4 times. The collection tube was replaced, and the sample was run through the Sep-Pack twice. The Sep-Pack was then washed 3 times with 1 mL of 0.1% FA before the peptides were eluted into a new collection tube via the addition of 500 μL 50% acetonitrile with 0.1% FA. This step was repeated to ensure that all peptides were eluted. The samples were then dried in a SpeedVac vacuum concentrator (John Morris Scientific, RVC 2-18) at 40 °C, with a vacuum of 40 mbar and a rotor speed of 1000 min^−1^. Finally, the samples were resuspended in 5 μL of 0.1% FA and used for subsequent MS data acquisition.

### 2.13. Proteomics Data Acquisition

LC-MS analysis was performed using an Ultimate 3000 RSLC nanosystem connected to an Orbitrap Exploris 480 mass spectrometer (Thermo Fisher Scientific, Bremen, Germeny). Peptides (1 μg) were resuspended in 0.1% formic acid and loaded onto a 25 cm fused silica column heated to 50 °C. The internal diameter (75 μm) of the column was packed with 1.9 um C18 particles. Peptide separation occurred over a 70 min linear gradient (3 to 20% acetonitrile in 0.1% formic acid) at a flow rate of 300 nL/min. The compensation voltages (−50 and −70 V) were applied from a FAIMS Pro interface (Thermo Fisher Scientific) to regulate the entry of ionised peptides into the mass spectrometer. The MS scans (*m/z* 300 to 1500) were acquired at a resolution of 60,000 (*m/z* 200) in positive ion mode. The MS/MS scans of fragment ions were measured at 15,000 resolution after the application of 27.5% HCD collision energy. A dynamic exclusion period of 40 s was specified. The mass spectrometry proteomics data have been deposited to the ProteomeXchange Consortium via the PRIDE [37] partner repository with the dataset identifier PXD034246.

### 2.14. Proteomics Data Analysis

The raw data was processed using the proteome discoverer platform (v2.4). The fragmentation Spectra were searched against the FASTA human database using the Sequest search engine with the precursor and fragment mass tolerance set to 10 ppm and 0.02 Da. Two missed cleavage sites were allowed, and the minimum peptide length was 6 amino acids. Oxidation and acetylation were included as variable modifications and carbamidomethylationwas included as a fixed modification.

Principle component analysis (PCA) was performed through the proteome discoverer platform using unscaled protein abundances. Hierarchical clustering was performed through the Proteome Discoverer platform using Euclidian distance function, and scaled before clustering.

### 2.15. Functional Annotation of Biological Process

Biological Process analysis was performed using the DAVID database [38]. The list of differentially abundant proteins was compared against the Gene Ontology-Biological Process (GO-BP) database, with a count threshold of 2 and EASE threshold of 0.1.

### 2.16. KEGG Global Metabolomic Network Analysis of Metabolites and Proteins of Interest

Network analysis was performed using the Metaboanalyst platform [34,35]. Proteins and metabolites with a differential abundant of at least 1.5-fold in CBPR vs. parental cell lines were investigated for related metabolomic networks in the KEGG Global Metabolomic Network, using the default settings.

### 2.17. Kaplan Meier Analysis

Proteins with a differential abundance of at least 1.5-fold in CBPR vs. parental cells in both OVCAR-5 and CaOV3 were selected for further investigation. The progression-free survival related to the selected genes was investigated in serous ovarian cancer, using the online Kaplan Meier (KM) plotter for the gene chip data of ovarian cancer [39,40]. The default settings were used, except for a restriction of analysis to patients with high-grade (grades 2 and 3) serous ovarian cancer who had received platinum-based chemotherapy, and the significance was deemed to be a *p*-value of less than 0.05.

## 3. Results

### 3.1. Generation and Growth Rate of CBPR Cells

Through the exposure of OVCAR-5 and CaOV3 cell lines to 6–8 cycles of CBP, we successfully generated two CBPR cell lines. The MTT viability assay was used to measure the growth rates of the cell lines over 72 h, and it showed no significant difference in growth rates between the parental and CBPR cell lines (Unpaired *t*-test, OVCAR-5: *p* = 0.331 and CaOV3: *p* = 0.818) (Figure 1A,B). MTT viability assays showed that the CBPR cells were significantly more resistant to CBP than the parental cell lines (OVCAR-5 parental IC50 = 88.6 μM, OVCAR-5 CBPR IC50 = 197.0 μM, *p* = 0.001; CaOV3 parental IC50 = 41.9 μM, CaOV3 CBPR IC50 = 124.0 μM, *p* = 0.02) (Figure 1C).

### 3.2. OVCAR-5 CBPR Cells Are More Motile than OVCAR-5 Parental In Vitro

An in vitro motility assay investigating the rate at which OVCAR-5 parental and CBPR cells move towards a chemoattractant (10% FBS) showed that the OVCAR-5 CBPR cells migrated at a significantly higher rate than the parental cell line (*p* = 0.037) (Figure 1D).

### 3.3. OVCAR-5 Cells Are More Invasive in the In Vivo CAM Assay

The CAM assay was performed with OVCAR-5 parental and OVCAR-5 CBPR cell lines to investigate their ability to invade the mesothelial layer in chicken embryos (Figure 1E,F). This model of cancer invasion replicates numerous aspects of the in vivo barriers to cancer cell metastasis, specifically the extracellular matrix (ECM). The CAM onplants were sectioned and stained to visualize OVCAR-5 cancer cell invasion, and the degree of invasion was measured as the total area of cancer cells (μm^2^) within the mesoderm of the CAM layer, using pan-cytokeratin immunohistochemistry. We observed that the OVCAR-5 CBPR cells were more invasive in vivo than the OVCAR-5 parental cells (Figure 1G, *p* = 0.0571).

### 3.4. LC-MSMS Analysis of Metabolites in Resistant vs. Parental Ovarian Cancer Cell Lines

The metabolomic profiles of the parental and CBPR pairs for both the OVCAR-5 and CaOV3 cell lines were analysed via LC-MS using a Thermo Scientific QExactive in biological triplicate. Analysis through the Metaboanalyst platform [34] putatively identified 380 metabolites in the OVCAR-5 samples (66 confirmed with reference standards) and 436 metabolites in the CaOV3 cell line samples (65 confirmed with reference standards) (OVCAR-5 parental: 369, OVCAR-5 CBPR: 370; CaOV3 parental: 436, CaOV3 CBPR: 428). The Venn diagrams show an almost complete overlap in the metabolites identified between the parental cell lines and their chemoresistant equivalents (Figure 2A,C). Volcano plots, calculated after filtering to remove cell culture media-related metabolites, show numerous metabolites that were differentially abundant between the parental and CBPR cells (1.5-fold, *p* < 0.05; more abundant in the OVCAR-5 parental: *n* = 29, less abundant in the OVCAR-5 parental: *n* = 7, more abundant in the CaOV3 parental: *n* = 29, less abundant in the CaOV3 parental: *n* = 11) (Figure 2B,D) (full details in Appendix A). Furthermore, the metabolic classes identified in these analyses favoured those that are related to amino acid metabolism and lipid metabolism (Figure 2E,F).

### 3.5. Classification of Ovarian Cancer Cell Lines Based on Metabolomic Profiles

The online platform, Metaboanalyst [34], was used to generate PCA plots and heat maps to investigate the separation of the parental and CBPR cell lines based on their molecular features. The PCA plots demonstrated a robust separation between the parental and CBPR cell lines based on their metabolomic profile (Figure 3A,C). This was further supported by hierarchical clustering and visualised in the heat maps (Figure 3B,D).

### 3.6. LC-MSMS Analysis of Proteins in Resistant vs. Parental Ovarian Cancer Cell Lines

The proteomic profiles of parental and CBPR pairs for both OVCAR-5 and CaOV3 cell lines were generated and analysed via LC-MSMS using the Exploris480 in biological triplicate, each of which was run as technical triplicates. An analysis with Proteome Discoverer resulted in over 5000 proteins being identified in each group (OVCAR-5 parental: 6423, OVCAR-5 CBPR: 6439; CaOV3 parental: 5395, CaOV3 CBPR: 5380). The Venn diagrams showed significant overlap in the proteins identified between both the parental cell lines and their chemoresistant equivalents (Figure 4A,C), while the volcano plots showed numerous proteins that were differentially abundant between the parental and CBPR cells (1.5-fold, *p* < 0.05; more abundant in OVCAR-5 parental: *n* = 154, more abundant in OVCAR-5 CBPR: *n* = 125, more abundant in CaOV3 parental: *n* = 124, more abundant in CaOV3 CBPR: *n* = 63) (Figure 4B,D). The full details of the proteomics results can be found in Appendix A.

### 3.7. Separation of Ovarian Cancer Cell Lines Based on Proteomic Profiles

The Proteome discoverer software was used to generate PCA plots and heat maps to investigate the separation of parental and CBPR cell lines based on their molecular features. The PCA plots demonstrated a robust separation between the parental and CBPR cell lines based on their proteomic profiles (Figure 5A,C). This was further supported by hierarchical clustering and was visualised in the heat maps (Figure 5B,D).

### 3.8. Functional Analysis of Differentially Abundant Proteins between Parental and CBPR Cancer Cell Lines

To assess the altered pathways observed between the parental and CBPR cell lines, we performed functional analysis on differentially abundant proteins using the Gene Ontology Biological Process database through DAVID. In the OVCAR-5 cells, there was a significant enrichment of structural biological processes (Table 1). In the CaOV3 cells, there was an enrichment of catabolic, survival and cell structure biological processes (Table 2).

Both OVCAR-5 and CaOV3 cell lines showed an enrichment of proteins with functions related to IFN1 signalling, response to cytokine and intermediate filament cytoskeletal organisation. There were no common differentially abundant proteins in the cell lines related to IFN1 signalling and response to cytokines, while keratin 2 and keratin 9 were both more abundant in OVCAR5 parental cells and less abundant in CaOV3 parental cells, and related to intermediate filament cytoskeletal organisation (Appendix A).

### 3.9. KEGG Global Metabolomic Network Analysis of Differentially Abundant Proteins and Metabolites between Parental and CBPR Cell Lines

To investigate how dysregulated proteins and metabolites contribute to altered metabolomic pathways in CBPR compared to parental ovarian cancer cell lines, we performed a KEGG Global Metabolomic Network analysis on differentially abundant proteins and metabolites using the Metaboanalyst platform [34]. In both cell lines, there was an enrichment of proteins with functions related to amino acid metabolism and energy metabolism-related pathways (Table 3 and Table 4).

Both the OVCAR-5 and CaOV3 cell lines showed significant enrichment in alanine, aspartate and glutamate metabolism, and in arginine and proline metabolism. One protein (glutamin-fructose-6-phosphate transaminase (isomerizing) 2 (GFPT2)) related to alanine, aspartate and glutamate metabolism was common to both the OVCAR-5 and CaOV3 cells. It was more abundant in the OVCAR-5 parental cells compared to the CBPR pair, and less abundant in the CaOV3 parental cells compared to the CBPR pair (Appendix A). GFPT2 was also seen to be more abundant in chemosensitive primary cells compared to chemoresistant cells (Appendix A).

### 3.10. Kaplan Meier Analysis of Proteins of Interest in Chemoresistance

To investigate whether the result of proteins that were differentially expressed in parental and CBPR cells was reflected in patients, we examined the relationship with PFS, and the gene expression of differentially abundant proteins present in both CaOV3 and OVCAR-5 CBPR cells compared to their chemosensitive counterparts, using the Kaplan Meir plotter (Appendix A). Further, we investigated proteins with differential abundances between the parental and CBPR cell lines that contributed to commonly identified metabolic pathways (Appendix A). We identified several proteins with differential expression between the CBPR and chemosensitive cell lines that were associated with patient survival (Figure 6).

RNA binding protein (PNO1) and mitogen-activated protein kinase 6 (MAPK6) both showed at least 1.5-fold expression in CBPR cells, and this increased expression was associated with significantly decreased PFS. High affinity copper uptake protein (SLC31A1) showed 1.5-fold higher expression in both OVCAR-5 and CaOV3 parental cells, and this increased expression was associated with significantly increased PFS. Asparagine synthase (ASNS) showed 1.5-fold higher expression in OVCAR-5 parental cells, is associated with alanine, aspartate and glutamate metabolism (Appendix A), and its increased expression was associated with significantly increased PFS in HGSOC patients.

## 4. Discussion

The molecular mechanisms defining chemosensitivity and chemoresistance remain poorly understood. A detailed understanding might enable us to predict treatment responses and may potentially lead to the identification of novel drug targets that aid therapy. The unbiased, comprehensive molecular characterisation that is achievable through mass spectrometry analysis has great potential for meeting this need and for providing deep insight into the molecular basis of chemoresistance. Here, we present a mass spectrometry analysis of cancer cell lines that were selected to exhibit resistance to treatment with the primary chemotherapeutic agent used for this disease, CBP. Mass spectrometry-based metabolomic and proteomic analysis of these samples allowed us to separate resistant cells from parental cells based on these respective molecular characteristics through unbiased statistical analysis. Further, dysregulated proteins and metabolites between resistant cells and their parental pairs were related to cancer-associated pathways.

There has been speculation regarding the cellular origins of the OVCAR5 cell line [26], and we include this information for transparency. However, it is speculative as to whether the differences observed in the phenotypical or molecular features reflect potentially different organs of origin.

While metabolomics has been widely employed in the search of HGSOC biomarkers (reviewed in [41]), there are relatively few studies utilising mass spectrometry-based metabolomics for the investigation of chemoresistance in HGSOC. One study investigated the metabolomic profiles of one ovarian cancer cell line, and its platinum-resistant derivative, and identified the impacts on methionine metabolism and glutathione synthesis pathways [42]. Here, we expanded on these findings, investigating two additional ovarian cancer cell lines and putatively identifying almost twice as many metabolites.

Furthermore, numerous proteomics investigations into chemoresistant ovarian cancer have been previously performed with the aim of understanding the molecular basis of chemoresistance and identifying biomarkers of chemoresistance to advise on treatment approaches [43]. These have employed multiple mass spectrometry techniques, including iTRAQ [44], ICAT [45,46], 2D DIGE coupled with MALDI-TOF-MS [47,48,49] or LC-MSMS [50], and label-free LC-MSMS [51]. In addition, mitochondrial fractionation has been employed to provide a deeper coverage of chemoresistance-associated proteins [48,51].

Previous analyses have identified pathways that correlate with a chemoresistant phenotype, such as glycolysis [44,50], ubiquitination [50], redox states [44], and PI3K signalling [45]. Furthermore, a broad panel of proteins has been highlighted for their potential as biomarkers of chemoresistant ovarian cancer (reviewed in [52]).

G. Fan, et al. (2015) [44] employed eight-plex iTRAQ MS to investigate 10 ovarian cancer cell lines, and identified several dysregulated proteins related to redox states and homologous repair in resistant cells. Interestingly, they were able to separate resistant from sensitive cell lines based on a panel of 300 differentially abundant proteins. Here, we demonstrate that a similar and potentially more robust separation can be achieved using over 6000 proteins when comparing directly between a cancer cell line and its resistant counterpart. Interestingly, Fan et al. were unable to achieve such a degree of separation with the DNA methylation and RNA data. A similar pattern has been observed previously [47], affirming the strength of proteomics-based approaches for the characterisation of chemoresistant phenotypes.

Exposure to chemotherapeutic agents has extreme impacts on the molecular and functional state of the affected cell. Chemoresistance is a major barrier to improving patient survival, not only because it prevents the clearance of the cancer, but because it often results in a more aggressive cancer upon relapse [53]. This could be a result of the evolutionary pressures which are placed upon cancer cells that are exposed to treatment, resulting in the development of adaptative survival and migration capabilities in response to these stressors. Our results show that repeated exposure to small doses of carboplatin is sufficient to produce an OVCAR-5 population that more readily migrates and invades, compared with its parent cell line, with very little difference in growth rates.

Molecular changes that occur in response to CBP treatment can be monitored in differentially abundant metabolites and proteins, and the biological function and pathways that they regulate. We applied SMPDB enrichment and Gene Ontology-Biological Process analysis to our metabolomic and proteomic data sets, respectively.

Most of the metabolites identified were not differentially abundant between the parental and chemoresistant cells. While the differences were sufficient to correctly cluster biological replicates, pathways analysis of dysregulated metabolites did not result in significant insights, and are summarised in Appendix A.

A manual analysis of dysregulated metabolites between parental and chemoresistant OVCAR-5 cells identified perturbations in glutaminolysis, the TCA cycle and glycolysis. The enrichment of glutaminolysis and the TCA cycle, was reflected by significant increases in glutamine (14-fold), oxoglutaric acid (2.7-fold), succinic acid (1.6-fold), malate (2.6-fold) and aspartic acid (1.9-fold), observed in OVCAR-5 CBPR cells compared to OVCAR-5 parental. Furthermore, we observed an increase in glycolysis-related metabolites in OVCAR5 CBPR, including pyruvic acid (7.2-fold) and glucose (3.4-fold). (Note that glutamine, aspartic acid and glucose were excluded from our above analysis (Table 3) due to their high abundances in the media-only control).

In contrast, a manual analysis of metabolites that were dysregulated between parental and chemoresistant CaOV3 cells did not show any clear trends. This could be attributed to very few of the dysregulated metabolites being confidently identified in this analysis (six metabolites were identified with reference standards out of 40 differentially abundant putative metabolites between the CaOV3 parental and CaOV3 CBPR cells (Figure 2)). Although it was difficult to interpret, the observation that several short peptides and some putatively identified glycosylated amino acids showed lower abundances in the CaOV3 CBPR cells (Appendix A), coupled with the observation from our proteomics data of enriched catabolic pathways in the CaOV3 CBPR cells (Table 1) potentially reflect alterations in protein turnover. Protein turnover and autophagy are well known to play complex roles in tumourogenesis and chemoresistance [54], although further studies are required to address their role in the development of CBP resistance.

Our manual analysis revealed only one metabolite that was similarly dysregulated in chemoresistance between the OVCAR5 and CaOV3 cell line pairs (Appendix A). Alanine was 2.8-fold higher in OVCAR5 parental and 2-fold higher in CaOV3 parental, compared to their CBPR pairs. While alanine is involved in several important metabolic pathways, including lactose metabolism [55], glycolysis and glucogenesis [56] and the alanine–glucose cycle [57], it is not clear how the increase in this metabolite contributes to the resistance phenotype in the absence of broader alterations in these related pathways.

No other metabolites were similarly dysregulated in chemoresistance between OVCAR5 and CaOV3 cell line pairs, suggesting that these cell lines possess distinct metabolomes. This could be due to differences in the metabolomes of the parental cells, as seen in Appendix A, or in differences in how the metabolism of each cell type responds to treatment, or a combination of the two.

Proteomic analysis of both cell line models revealed enriched biological processes that were related to response to cytokines and cellular response to IFN1 (Table 1 and Table 2). IFN1 production and signalling has been previously implicated in the chemotherapy response of neoplastic cells [58]. However, chemoresistance has previously been associated with a decreased expression of IFN1 genes in ovarian cancer, contributing to an immunosuppressed microenvironment [59]. Moreover, A. Sistigu, et al. (2014) [60] showed that cisplatin treatment of a panel of cell lines was unable to stimulate IFN1 gene expression. However, they demonstrated that supplying exogenous IFN1 enhanced the anti-neoplastic effects of carboplatin in a mice model of melanoma [60]. The role of IFN signalling in ovarian cancer chemoresistance merits further investigation. Combination therapy with IFN-gamma, Carboplatin and Paclitaxel for the treatment of ovarian cancer has previously been investigated in a phase III clinical trial [61]. Combining diverse immunotherapy approaches with standard chemotherapy has also shown promise in treating ovarian cancer [62]. Our findings merit further investigation into combination therapy with cytokines and carboplatin, with the express aim of overcoming chemoresistance. Further investigations will be necessary to better understand the molecular details and biological relevance of these findings.

Additional proteins are implicated in the regulation of IFN signalling. For example, our results showed a decreased abundance of tyrosine kinase 2 (TYK2) in CaOV3 CBPR cells, which plays a key role in the promotion of IFN1 signalling in response to cytokines (Appendix A). Moreover, we observed an increased abundance of proteosome subunit beta 8 (PSMB8) in OVCAR-5 CBPR cells (Appendix A). This protein can be activated by IFN signalling to form the ‘immunoproteosome’ [63]. In addition to antigen processing, this complex promotes cell survival, and its inhibition has been shown to sensitize drug-resistant stomach and colon cancer cell lines to cisplatin [64]. Interestingly, the stimulation of IFN1 genes is currently under investigation as a potential therapeutic in a range of contexts and cancers [59,65].

The metabolic network analysis of dysregulated metabolites and proteins, using the KEGG Global Metabolomic Pathways database, identified ‘alanine, aspartate and glutamate metabolism’ and ‘arginine and proline’ metabolism as enriched pathways in both OVCAR-5 and CaOV3 cells. Auxotrophy, where the cell relies on external sources for a specific molecule, for alanine and glutamate, have both been observed in cancer [66,67], and it is theorized that this can promote growth and survival through exogenous amino acid importation pathways [66]. In our results, we observed an increased abundance of alanine in CBPR for both cell lines, while glutamate showed an increased abundance in OVCAR5 CBPR, but a decreased abundance in CaOV3 CBPR (Appendix A). There is a significant demand for glutamate in proliferating cells for transamination reactions, and for the use of its carbon backbone for the synthesis of other anabolic metabolites and antioxidants [68]. Further, glutamate contributes to the TCA cycle (it was enriched according to a KEGG global metabolomic pathway analysis for OVCAR-5 (Table 3)) through the process of ‘glutaminolysis’, which contributes to energy production and cell survival [69]. However, due to the differences between our cell line pairs, it is difficult to draw broad and consistent, conclusions regarding how the metabolome changes in response to acquired resistance to CBP.

A deeper investigation into the enrichment of the ‘arginine and proline’ pathway revealed the contribution of proteins and metabolites involved in the creatine (creatine kinase (CK) and guanidoacetic acid) and urea cycle (arginase and 4-guanidinobutanoic acid) (Appendix A). The creatine pathway is responsible for maintaining energy homeostasis through the reversible biosynthesis of phosphocreatine (PCr) from ATP [70]. The presence of a pool of PCr facilitates the rapid generation of ATP at sites with high energy demands without the need for transporting ATP across cellular membranes [71]. As cell division is regulated in an energy-dependent manner [72], CK-regulated ATP homeostasis is important for the progression of the G1 and G2 phases into S phase and M phase, respectively [73]. The urea cycle is often dysregulated in cancer to maximize the available nitrogen and carbon for the anabolic synthesis of macromolecules that are required for rapid tumour proliferation and growth [74]. Further, the urea cycle is essential for the detoxification of ammonia, through its conversion to urea [75], which accumulates primarily as a byproduct of the glutaminolysis reaction [74], which was seen to be enriched in OVCAR5 CBPR cells, as discussed above. These altered pathways potentially represent perturbations in energy homeostasis, cell cycle control, anabolic pathways, and the detoxification of anabolic waste products. However, the relatively small number of metabolites and proteins that were observed to contribute to these pathways in this study means that these results must be interpreted with caution. Further, the observation that the two cell lines employed in this study have distinct metabolic pathways, both at base line and after acquired resistance to CBP, limit the broader conclusions that can be drawn from this data.

To expand on the proteins identified as being differentially abundant between both the CBPR and chemosensitive cancer cell lines, we investigated the relationship between their expression and patient survival, using a KM plotter [39]. We identified two proteins (RNA binding protein PNO1 (PNO1) and mitogen-activated protein kinase 6 (MAPK6)) that exhibited increased abundance in CBPR cells and decreased PFS in HGSOC patients when their related gene expression was high.

PNO1 is involved in ribosomal biogenesis, and its knockdown has been seen to increase p53 and p21 signalling, resulting in apoptosis in colon cancer cell lines [76]. In addition, its expression has been related to lung adenocarcinoma progression mediated by amplified Notch signalling pathways [77]. Downregulation of the Notch signalling pathway sensitizes lung cancer cells to cisplatin [78], potentially representing a pathway through which PNO1 expression promotes resistance to platinum-based chemotherapy.

MAPK6 is a multifunctional signalling protein that is involved in inflammatory responses, and cell growth and differentiation [79]. It has been reported to promote metastasis in lung cancer [80] and inhibit apoptosis in HUVEC cells [81]. It also phosphorylates tyrosyl DNA phosphodiesterase 2 (TDP2), promoting its topisomerase 2-linked DNA repair mechanism, which has been shown to confer resistance to topoisomerase inhibitors [82].

We also identified the high affinity copper transporter protein (SLC31A1), which showed decreased abundance in CBPR cell lines and increased PFS related to high *SLC31A1* expression. SLC31A1 is the main import channel for platinum-based antineoplastic drugs into the cell [83], and its increased expression confers platinum sensitivity in several contexts [84,85]. Increased SLC31A1 expression, in combination with platinum-based chemotherapy, correlates with increased survival in ovarian [86] and lung cancer [87]. As SLC31A1 expression increases in response to limited copper availability, there have been numerous clinical trials investigating the use of copper-chelating agents to increase SLC31A1 expression and to subsequently sensitize cells to platinum-based chemotherapy [88,89,90].

Further, we identified a protein (asparagine synthase (ASNS)) that was upregulated in OVCAR-5 parental cells, related to the enriched ‘alanine, aspartate and glutamate metabolism’ metabolic network, and whose high expression related to increased PFS in HGSOC patients. This protein catalyses the synthesis of asparagine from aspartate and glutamate [91], playing a role in tumour initiation and growth under amino acid-limiting conditions [92]. Silencing ASNS in nasopharyngeal carcinoma has been demonstrated to sensitize cells to cisplatin treatment through impaired DNA repair and cell survival mechanisms [93]. Interestingly, supplementation with exogenous asparagine improved the growth rates of the tumour cells, but did not impact on resistance to cisplatin [93]. In contrast, increased ASNS expression in response to glucose starvation was seen to enhance cisplatin resistance in pancreatic cancer [94]. Further work is required to understand how ASNS contributes to platinum resistance in ovarian cancer, but in acute lymphoblastic leukaemia, in which the *ASNS* gene is silenced, asparaginase treatment is effective at limiting tumour growth, representing a metabolism-targeted treatment in ASNS deficient cells [95].

It is important to note that we showed these KM curves to demonstrate that these proteins of interest may play a role in disease progression in a larger cohort. However, as these were only a small portion of the up- and downregulated proteins identified, we do not imply that that these are the main regulators of chemoresistance in this setting. While the roles of these proteins in chemoresistance do merit further investigation, we have not pursued further validation experiments (such as Western blots, knockdown, or overexpression experiments).

We have been successful in identifying numerous pathways that are altered in chemotherapy-resistant cells lines. However, it is difficult to draw broader conclusions, due to our relatively small sample size and the observation that PCA and the hierarchical clustering of metabolomic and proteomic data favour clustering that is based on cell type rather than chemoresistance status (Appendix A). Furthermore, we accept that the in vitro exposure of these cells to chemotherapy, in the absence of an in vivo microenvironment, results in a chemotherapy-resistant phenotype that does not accurately reflect the in vivo situation. Our model also does not replicate chemoresistance mechanisms that exist outside of the cell, such as extracellular signalling, drug exclusion and the influence of other features, including hypoxia and nutrient deprivation.

A proteomic analysis of patient-derived samples is required in order to better understand the molecular features of chemoresistance in vivo. Here, we performed a pilot proteomic analysis on two patient samples taken from the ascites of one patient with a chemosensitive disease and one with a chemoresistant disease (Appendix A). Our findings highlight that there is a striking difference in the molecular profiles of the cell lines and primary cells that is far more significant than the differences attributed to chemosensitivity status (Appendix A). Of interest is the observed enrichment of responses to cytokines and cellular responses to IFN1 biological processes in relation to differentially abundant proteins between sensitive and resistant primary cells (Appendix A), which is also observed in the cell line proteomics data. Currently, we have access to only two well-characterized and matched primary samples for analysis, but these findings provide a foundation to expand upon this study using a larger cohort of primary samples in the future.

A previous study investigated the molecular profiles of ascites-derived ovarian cancer cells in a larger cohort [96]. They were able to identify approximately 2800 proteins across four sensitive and four resistant samples, using SDS-PAGE protein separation followed by mass spectrometry analysis using an Orbitrap Elite mass spectrometer (Thermo Fisher Scientific, Adelaide, SA, Australia). From these proteins, they identified a total of 353 differentially abundant proteins between the resistant and sensitive groups, and observed enriched metabolic, DNA repair and host immune response pathways in resistant cells. Interestingly, they identified the ‘spheroid’ structures as being essential in ovarian cancer chemoresistance, representing a structural feature of ovarian cancer that is not captured in traditional cell culture.

With the application of the modern mass spectrometry approaches outlined here to a larger cohort of patient-derived samples, there is the potential to further develop our understanding of chemoresistance. Furthermore, the application of advanced in vitro cell culturing techniques, including ovarian cancer spheroids, promises to help bridge the gap between the in vitro and in vivo settings. Finally, combining multiple molecular analyses beyond metabolomics and proteomics, in a ‘panomics’ approach [97] promises to provide a deeper understanding of the molecular underpinnings of chemoresistance, as has recently been demonstrated in a study of low-grade serous ovarian cancer [98]. Together, these advances hold the potential to provide a holistic molecular snapshot of chemoresistance in a biologically and clinically relevant manner, to improve patient outcomes.

## 5. Conclusions

Our analysis was able to separate chemoresistant cells from their parental cells based on their metabolomic and proteomic features, and we identified altered biological processes and pathways that are of further interest. A preliminary investigation of patient-derived cells highlighted the need to perform broad biological and molecular analyses, and comprehensive in vitro and in vivo studies using a larger patient cohort, to achieve a deeper and more clinically relevant characterisation of the molecular drivers of chemoresistance.

## Figures and Tables

**Figure 1 cancers-14-02763-f001:**
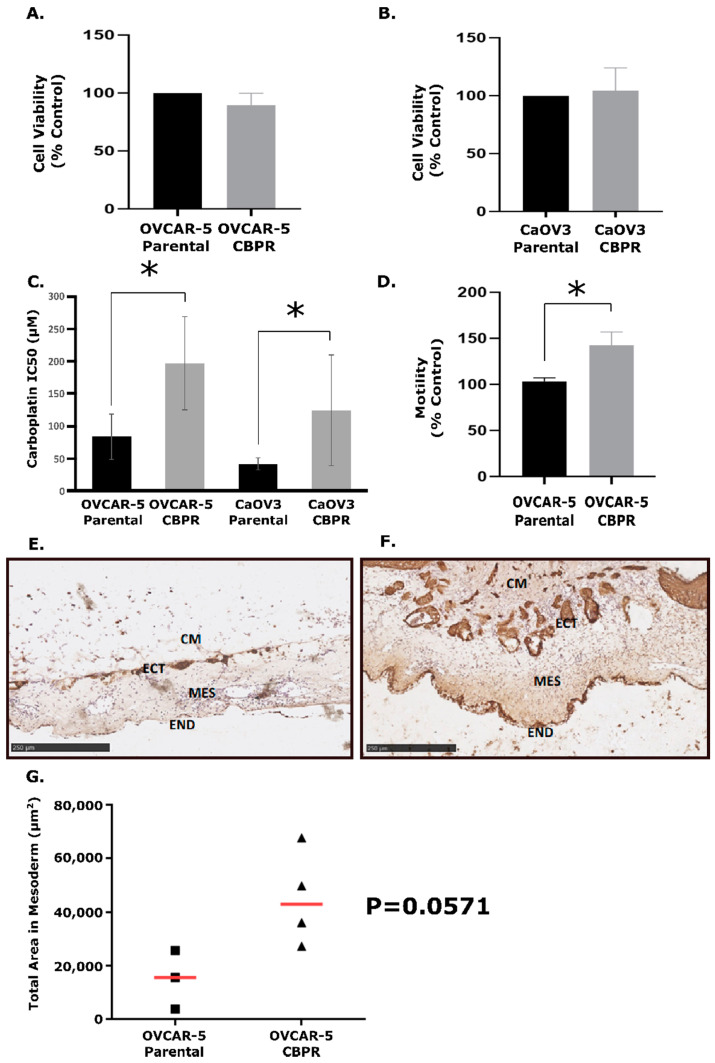
Biological data for carboplatin-resistant cancer cell lines. Growth rates determined using an MTT assay of (**A**) OVCAR-5 parental vs. CBPR and (**B**) CaOV3 parental vs. CBPR show no significant difference (unpaired *t*-test, OVCAR-5: *p* = 0.331 and CaOV3: *p* = 0.818). (**C**) Dose response of OVCAR-5 and CaOV3 resistant and parental pairs to carboplatin. OVCAR-5 parental IC50 = 83.6 (*n* = 8), OVCAR-5 CBPR IC50 = 1967.0 (*n* = 8); CaOV3 parental IC50 = 41.9 (*n* = 7), CaOV3 CBPR IC50 = 124.0 (*n* = 7). Unpaired T test showed IC50 to be significantly higher in CBPR cell lines compared to their parental cells (OVCAR-5 *p* = 0.001, CaOV3 *p* = 0.02). (**D**) In vitro motility assay of OVCAR-5 parental compared to OVCAR-5 CBPR cell lines showed that OVCAR-5 CBPR is significantly more motile than its parental pair (unpaired *t*-test, *p* = 0.037). In vivo CAM invasion assay with OVCAR-5 parental (**E**) and OVCAR-5 CBPR (**F**). OVCAR-5 cell Matrigel grafts (CM) were placed in the top of the ectoderm (ECT) layer and cancer cell invasion into the CAM mesoderm (MES) layers was assessed on Day 14 of chick embryo development. END = endoderm CAM paraffin sections (6 µm) were immunostained with pan-cytokeratin antibody. Scale bar = 250 µm. (**G**) Quantification of OVCAR-5 parental (*n* = 3) and OVCAR-5 CBPR (*n* = 4) invasion into the CAM mesoderm. Data represents the total pan-cytokeratin positive area (µm^2^) in the mesoderm area from 5 to 6 images per embryo. Mann–Whitney U test showed greater invasion in OVCAR-5 CBPR compared to parental cells (*p* = 0.0571) (* = statistical significance, *p* < 0.05).

**Figure 2 cancers-14-02763-f002:**
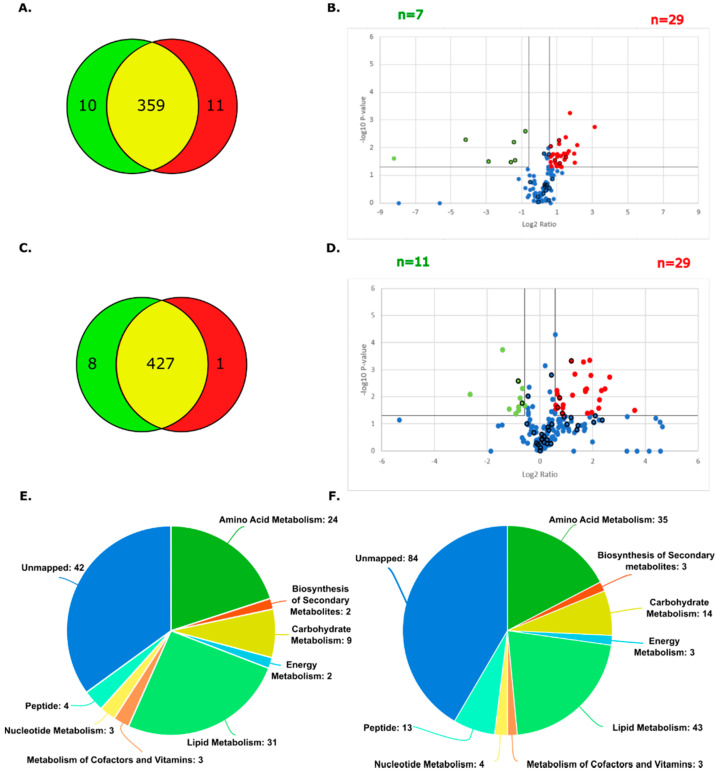
Summary of metabolites identified in parental and CBPR cell lines. (**A**) Metabolites putatively identified in OVCAR-5 cell line pair. OVCAR-5 parental total= 369, OVCAR-5 CBPR total= 370, unique to OVCAR-5 parental= 10, unique to OVCAR-5 CBPR= 1, in both OVCAR-5 parental and CBPR= 359. (**B**) Volcano plot showing metabolites with 1.5-fold (=0.58 log2) differential abundance in OVCAR-5 parental compared to CBPR. Twenty-nine metabolites (5 identified with reference standards indicated by bold outline) were upregulated, and 7 metabolites (6 identified with reference standards indicated by bold outline) were downregulated in parental OVCAR-5 cells. (**C**) Metabolites identified in CaOV3 cell line pair before filtering: CaOV3 parental total= 435, CaOV3 CBPR total= 428, unique to CaOV3 parental = 8, unique to CaOV3 CBPR= 1, in both CaOV3 parental and CBPR= 436. (**D**) Volcano plot showing metabolites with 1.5-fold (=0.58 log2) differential abundance in CaOV3 parental compared to CBPR. Twenty-nine metabolites were seen to be upregulated (4 identified with reference standards indicated by bold outline) and 11 metabolites (2 identified with reference standards indicated by bold outline) were downregulated in parental CaOV3 cells. (**E**) Pie chart of metabolite classes identified in OVCAR-5 parental and CBPR cell lines. (**F**) Pie chart of metabolite classes identified in CaOV3 parental and CBPR cell lines.

**Figure 3 cancers-14-02763-f003:**
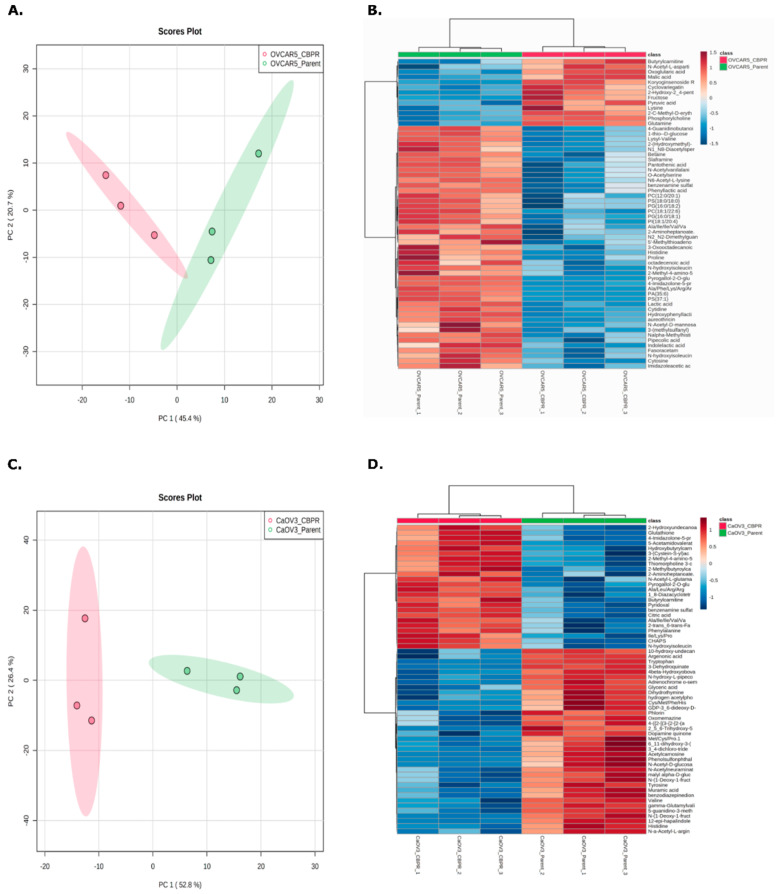
Principle component analysis (PCA) and hierarchical clustering of top 50 differentially abundant metabolites (identified through *t*-test) between parental (green) and the CBPR (red) cancer cell lines (**A**,**B**) OVCAR-5 and (**C**,**D**) CaOV3. Analysis was performed with Metaboanalyst.

**Figure 4 cancers-14-02763-f004:**
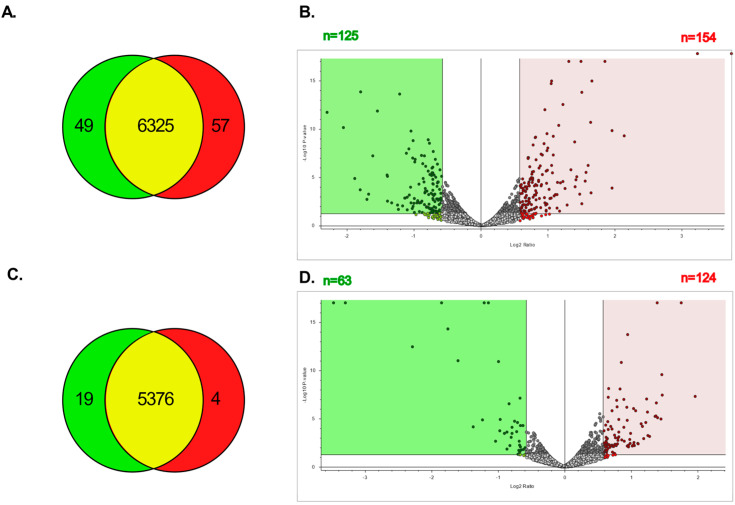
Total number of proteins identified in parental and CBPR cell lines. (**A**) Proteins identified in OVCAR-5 cell line pair: OVCAR-5 parental total = 6374, OVCAR-5 CBPR total = 6382, OVCAR-5 parental exclusive = 49, OVCAR-5 CBPR exclusive = 57, in both OVCAR-5 parental and CBPR = 6325. (**B**) Volcano plot showing proteins with 1.5-fold (=0.58 log2) differential abundance in OVCAR-5 parental compared to CBPR. A total of 154 proteins were upregulated and 125 proteins were downregulated in parental OVCAR-5 cells. (**C**) Proteins identified in CaOV3 cell line pair: CaOV3 parental total = 5395, CaOV3 CBPR total = 5380, CaOV3 parental exclusive = 19, CaOV3 CBPR exclusive = 4, in both CaOV3 parental and CBPR: 5399. (**D**) Volcano plot showing proteins 1.5-fold (=0.58 log2) differentially abundant in CaOV3 parental compared to CBPR. A total of 124 proteins were seen to be upregulated and 63 proteins were downregulated in parental CaOV3 cells.

**Figure 5 cancers-14-02763-f005:**
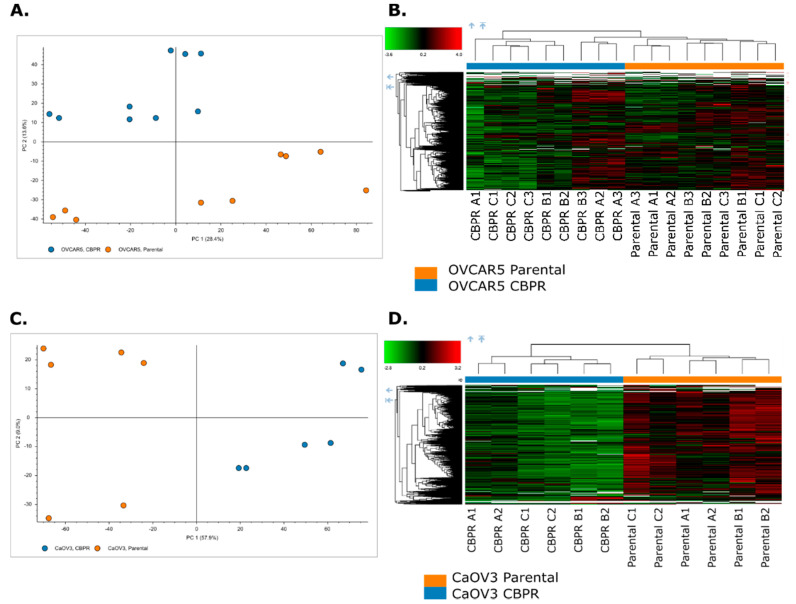
Principle component analysis (PCA) and hierarchical clustering of proteomic features of parental (orange) and the CBPR (blue) cancer cell lines (**A**,**B**) OVCAR-5, (**C**,**D**) CaOV3. Analysis was performed with Proteome Discoverer (Thermo) inbuilt data visualisation tools.

**Figure 6 cancers-14-02763-f006:**
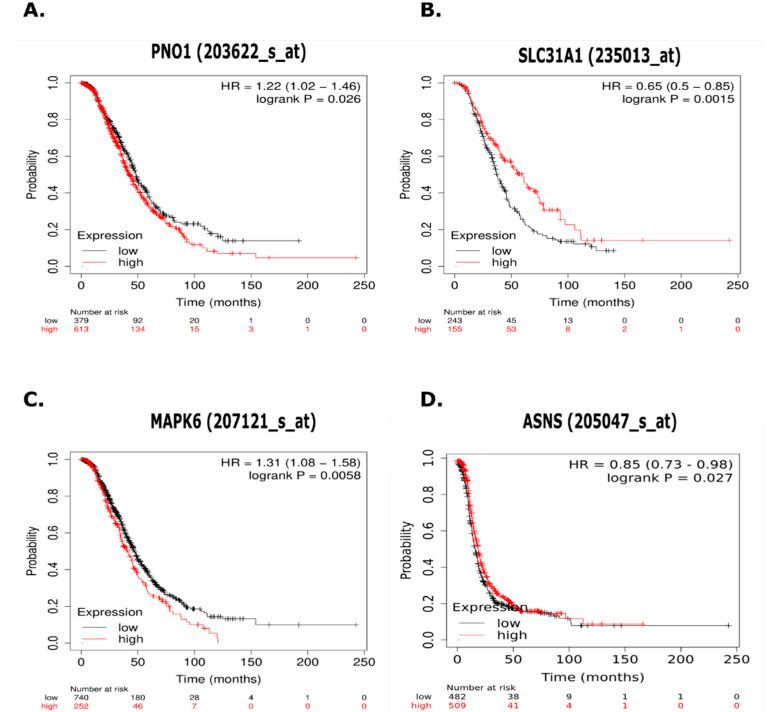
Kaplan Meier (KM) plots measuring progression free survival (PFS) in serous ovarian cancer patients treated with platinum-based chemotherapy (*n* = 979) related to selected proteins. (**A**) RNA binding protein *PNO1* (*PNO1*) showed abundance at least 1.5-fold lower in parental cell lines and is associated with significantly decreased PFS. (**B**) High affinity copper uptake protein (*SLC31A1*) showed abundance at least 1.5-fold higher in parental cell lines and is associated with significantly increased PFS. (**C**) Mitogen-activated protein kinase 6 (*MAPK6)* showed abundance at least 1.5-lower in parental cell lines and is associated with significantly decreased PFS. (**D**) Asparagine synthase (ASNS) showed abundance at least 1.5-fold higher in OVCAR-5 parental cells, is part of the enriched Alanine, Aspartate and Glutamate Metabolism pathway, and is associated with significantly increased PFS.

**Table 1 cancers-14-02763-t001:** Top 10 gene ontology biological functions for proteins with decreased or increased abundance in OVCAR-5 cells. Terms in bold represent those which are represented in both OVCAR-5 and CaOV3 cells.

Rank	Term	Count	Involved Genes/Total Genes (%)	*p*-Value
1	cytoskeleton organisation	29	13.1	3.2 × 10^−0.4^
2	antigen processing and presentation of peptide antigen	10	4.5	3.5 × 10^−0.4^
3	cellular component assembly	51	23	6.3 × 10^−0.4^
4	**response to cytokine**	22	9.9	8.1 × 10^−0.4^
5	cell junction organisation	11	5	9.8 × 10^−0.4^
6	cytokine-mediated signalling pathway	17	7.7	1.3 × 10^−0.3^
7	**intermediate filament cytoskeleton organisation**	5	2.3	1.7 × 10^−0.3^
8	regulation of cellular component organisation	45	20.3	1.7 × 10^−0.3^
9	**type I interferon signalling pathway**	6	2.7	2.1 × 10^−0.3^
10	cell junction assembly	9	4.1	2.5 × 10^−0.3^

**Table 2 cancers-14-02763-t002:** Top 10 gene ontology biological functions for proteins with decreased or increased abundances in CaOV3 cells. Terms in bold represent those which are represented in both OVCAR-5 and CaOV3 cells.

Rank	Term	Count	Involved Genes/Total Genes (%)	*p*-Value
1	negative regulation of necroptotic process	3	2.2	1.8 × 10^−0.3^
2	**response to type I interferon**	5	3.7	2.5 × 10^−0.3^
3	cellular macromolecule catabolic process	16	11.9	4.8 × 10^−0.3^
4	negative regulation of cellular protein metabolic process	16	11.9	5.5 × 10^−0.3^
5	protein catabolic process	14	10.4	6.1 × 10^−0.3^
6	positive regulation of extrinsic apoptotic signalling pathway	4	3	7.1 × 10^−0.3^
7	**intermediate filament organisation**	3	2.2	1.0 × 10^−0.2^
8	**response to cytokine**	13	9.7	1.4 × 10^−0.2^
9	regulation of protein ubiquitination	7	5.2	1.0 × 10^−0.2^
10	positive regulation of proteolysis	8	6	1.60 × 10^−0.2^

**Table 3 cancers-14-02763-t003:** Top 10 KEGG Global Metabolic Pathways for proteins and metabolites with decreased or increased abundances in OVCAR-5 cells. Terms in bold represent those which are represented in both OVCAR-5 and CaOV3 cells.

Rank	Metabolite Set	Count (Metabolites)	Count (Proteins)	Count (Total)	*p*-Value
1	**Alanine, aspartate and glutamate metabolism**	4	2	6	0.0000817
2	Glycolysis/Gluconeogenesis	2	3	5	0.000717
3	Pyruvate metabolism	2	3	5	0.00419
4	Inositol phosphate metabolism	0	4	4	0.0053
5	**Arginine and proline metabolism**	3	1	4	0.0189
6	Citrate cycle (TCA cycle)	2	1	3	0.0208
7	Limonene and pinene degradation	1	0	1	0.03
8	Chloroalkane and chloroalkene degradation	1	1	2	0.0348
9	Valine, leucine and isoleucine degradation	2	1	3	0.0445
10	Fatty acid biosynthesis	0	2	2	0.0452

**Table 4 cancers-14-02763-t004:** Top 10 KEGG Global Metabolic Pathways for proteins and metabolites with decreased or increased abundances in CaOV3 cells. Terms in bold represent those which are represented in both OVCAR-5 and CaOV3 cells.

Rank	Metabolite Set	Count (Metabolites)	Count (Proteins)	Count (Total)	*p*-Value
1	**Alanine, aspartate and glutamate metabolism**	2	1	3	0.00692
2	**Arginine and proline metabolism**	1	2	3	0.0148
3	Folate biosynthesis	0	2	2	0.045
4	Linoleic acid metabolism	0	1	1	0.0858
5	Vitamin B6 metabolism	1	0	1	0.0893
6	Glycine, serine and threonine metabolism	2	0	2	0.0895
7	Glycosylphosphatidylinositol (GPI)-anchor biosynthesis	0	1	1	0.134
8	Thiamine metabolism	0	1	1	0.145
9	Amino sugar and nucleotide sugar metabolism	0	2	2	0.145
10	Sphingolipid metabolism	0	1	1	0.181

## Data Availability

The data presented in this study are available in this article, referenced online data depositories and Appendix A.

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
