# Peer review of "Chemoresistant Cancer Cell Lines Are Characterized by Migratory, Amino Acid Metabolism, Protein Catabolism and IFN1 Signalling Perturbations"

_cancers, 2022, doi:10.3390/cancers14112763_

Round 1

Reviewer 1 Report

In the article, the authors represent carboplatin-resistant cell line models using OVCAR5 and CaOV3 cell lines to identify chemoresistance-specific molecular features. They also tested 2 patient samples. Authors are encouraged to discuss the potential diagnostic, prognostic, and therapeutic meaning, utility value, and limitations of the study. Please explain why only 2 patients were included in the study.

Authors are encouraged to mention related guidelines (like NCCN, ESMO, some national guidelines, etc.) to support the statement in lines 43-44.

According to the statement in line 87 “Recent reports indicate that OVAR-5 might originate from metastatic gastrointestinal cancer and were potentially wrong fully labelled as Ovarian cancer [21].”, authors are encouraged to explain why they use this cell line in this study. The difference represented in Figure 2, A-D could be because of the different cell line origin (“gastro vs ovary”)

In line 210 abbreviation GO-BP should be explained. Please check for any other abbreviations.

Pictures could be more self-explainable (additional comments, arrows, etc). All pictures should have the same styling, font – different fonts are confusing and difficult to understand. Some pictures could be merged into one figure (for example Figures 1A and 1B).

In Table 1 and 2 authors should explain how “proteins with decreased or increased abundance” are marked in the Tables. Also in Supplementary Table 2, “sheet4” does not have a “title” and the table header is missing.

In the results, some comparisons of cell lines vs CBPR cell lines are not presented. For example Results in paragraphs, 3.2 and 3.3, Figure 1D, E, F, and G - cell line CaCOV3 comparison of origin vs CBPR is missing. Authors are encouraged to present all comparison analysis results between OVCAR-5 origin vs CBPR and CaOV3 origin vs CBPR. The same type of results could be presented in one graph.

Authors are encouraged to state if both patients had signed the informed consent and to send the informed consent.

I am looking forward to reading the improved case report.

Author Response

In the article, the authors represent carboplatin-resistant cell line models using OVCAR5 and CaOV3 cell lines to identify chemoresistance-specific molecular features. They also tested 2 patient samples. Authors are encouraged to discuss the potential diagnostic, prognostic, and therapeutic meaning, utility value, and limitations of the study. Please explain why only 2 patients were included in the study.

Explain in line 728 that we investigated these primary samples as complimentary data. Currently, we only have access to these two primary samples but aim to expand on this cohort in future studies.

Authors are encouraged to mention related guidelines (like NCCN, ESMO, some national guidelines, etc.) to support the statement in lines 43-44.

Addressed by including reference to the 2021 NCCN guidelines for ovarian cancer treatment which recommend treatment with platinum based chemothereapy and taxane based chemotherapy for all stage II, III and IV ovarian cancers following initial surgery (Line 46)

According to the statement in line 87 “Recent reports indicate that OVAR-5 might originate from metastatic gastrointestinal cancer and were potentially wrong fully labelled as Ovarian cancer [21].”, authors are encouraged to explain why they use this cell line in this study. The difference represented in Figure 2, A-D could be because of the different cell line origin (“gastro vs ovary”)

While there is speculation about the origins of OVCAR5 (which we have included for transparency) it is only speculative that the differences observed between cell lines is due to this. We expand on this in the manuscript in line 516

In line 210 abbreviation GO-BP should be explained. Please check for any other abbreviations.

Provided the full title of this abbreviation in line 287

Pictures could be more self-explainable (additional comments, arrows, etc). All pictures should have the same styling, font – different fonts are confusing and difficult to understand. Some pictures could be merged into one figure (for example Figures 1A and 1B).

Addressed by reworking Figure 1 to have consistent styling and fonts.

In Table 1 and 2 authors should explain how “proteins with decreased or increased abundance” are marked in the Tables. Also in Supplementary Table 2, “sheet4” does not have a “title” and the table header is missing.

Both up and down regulated proteins are included in this analysis to identify pathways which may be dysregulated in chemoresistance. We do not claim that our data is sufficient to identify exactly how these complex pathways are dysregulated but rather identify areas for further investigation. For this reason we chose not to include this information here.

‘Sheet 4’ in  Supplementary Table 2 was left in by accident. All data contained within that sheet is also contained with the sheet: ‘Primary Cells’

In the results, some comparisons of cell lines vs CBPR cell lines are not presented. For example Results in paragraphs, 3.2 and 3.3, Figure 1D, E, F, and G - cell line CaCOV3 comparison of origin vs CBPR is missing. Authors are encouraged to present all comparison analysis results between OVCAR-5 origin vs CBPR and CaOV3 origin vs CBPR. The same type of results could be presented in one graph.

Cell mobility and CAM assay data are unavailable for CaOV3 cells at this time 

Authors are encouraged to state if both patients had signed the informed consent and to send the informed consent.

We have provided the ethics approval for the project from which the primary samples were gathered. Samples were collected with informed consent

Reviewer 2 Report

The work is a significant contribution to the research on ovarian cancer.

I recommend that in vivo studies involving a larger cohort should be carried out.

Some minor corrections were observed: Line 48,  68 and 163. In line 48 - ROS was rendered as reactive oxygen series instead of reactive oxygen species. Line 68 and 163 had a typographical error.

Author Response

The work is a significant contribution to the research on ovarian cancer.

I recommend that in vivo studies involving a larger cohort should be carried out.

Explain in line 728 that we investigated these primary samples as complimentary data. Currently, we only have access to these two primary samples but aim to expand on this cohort in future studies.

Some minor corrections were observed: Line 48,  68 and 163. In line 48 - ROS was rendered as reactive oxygen series instead of reactive oxygen species. Line 68 and 163 had a typographical error.

This is corrected in line 52

Reviewer 3 Report

The article Acland et al is devoted to carboplatin resistant in ovarian cancer. Authors developed two carboplatin resistant cell lines: OVCAR5 and CaOV3. Using mass spectrometry analysis the proteome of these cell lines was analyzed.  

However there are several comments and suggestions:

  1. It will be important to show the level of expression RNA binding protein (PNO1), mitogen-activated protein kinase 6 (MAPK6), (SLC31A1) and lactotransferrin (LFT) in resistant cell lines OVCAR5 and CaOV3 by western-blot analysis.
  2. It will be important also to study the effects of these proteins PNO1, MAPK6, SLC31A1, LFT in resistant cell lines OVCAR5 and CaOV3 using knockdown or overexpression.
  3. Very exciting result is that resistant cell lines exhibited processes related to response to cytokines and cellular response to IFN1. May be it will be interesting to study the effectives of co-treatment of cytokines and carboplatin of OVCAR5 and CaOV3 cell lines   

Author Response

However there are several comments and suggestions:

  1. It will be important to show the level of expression RNA binding protein (PNO1), mitogen-activated protein kinase 6 (MAPK6), (SLC31A1) and lactotransferrin (LFT) in resistant cell lines OVCAR5 and CaOV3 by western-blot analysis.

In line 702 we explain that, while we have expanded on proteins of interest, we do not claim that the proteins identified are the key regulators of chemoresistance in this setting. For this reason we have not pursued further validation experiments.

  1. It will be important also to study the effects of these proteins PNO1, MAPK6, SLC31A1, LFT in resistant cell lines OVCAR5 and CaOV3 using knockdown or overexpression.

In line 702 we explain that, while we have expanded on proteins of interest, we do not claim that the proteins identified are the key regulators of chemoresistance in this setting. For this reason we have not pursued further validation experiments.

  1. Very exciting result is that resistant cell lines exhibited processes related to response to cytokines and cellular response to IFN1. May be it will be interesting to study the effectives of co-treatment of cytokines and carboplatin of OVCAR5 and CaOV3 cell lines   

Provided some further information about the investigations of combination therapy with cytokines and carboplatin in line 607.